# PCSK9 Imperceptibly Affects Chemokine Receptor Expression In Vitro and In Vivo

**DOI:** 10.3390/ijms222313026

**Published:** 2021-12-01

**Authors:** Sai Sahana Sundararaman, Linsey J. F. Peters, Sumra Nazir, Andrea Bonnin Marquez, Janneke E. Bouma, Soyolmaa Bayasgalan, Yvonne Döring, Emiel P. C. van der Vorst

**Affiliations:** 1Interdisciplinary Centre for Clinical Research (IZKF), RWTH Aachen University, 52074 Aachen, Germany; ssahanasunda@ukaachen.de (S.S.S.); lipeters@ukaachen.de (L.J.F.P.); snazir@ukaachen.de (S.N.); anbonninmarq@ukaachen.de (A.B.M.); jbouma@ukaachen.de (J.E.B.); 2Institute for Molecular Cardiovascular Research (IMCAR), RWTH Aachen University, 52074 Aachen, Germany; 3Maastricht University Medical Centre, Department of Pathology, Cardiovascular Research Institute Maastricht (CARIM), 6229 ER Maastricht, The Netherlands; 4Institute for Cardiovascular Prevention (IPEK), Ludwig-Maximilians-University Munich, 80336 Munich, Germany; Soyolmaa.Bayasgalan@med.uni-muenchen.de; 5German Centre for Cardiovascular Research (DZHK), Partner Site Munich Heart Alliance, 80336 Munich, Germany; 6Swiss Cardiovascular Center, Division of Angiology, Inselspital, Bern University Hospital, University of Bern, CH-3010 Bern, Switzerland

**Keywords:** PCSK9, hematopoietic cells, vascular inflammation, chemokine receptors, smooth muscle cells, endothelial cells

## Abstract

Proprotein convertase subtilin/kexin type 9 (PCSK9) is a protease secreted mainly by hepatocytes and in lesser quantities by intestines, pancreas, and vascular cells. Over the years, this protease has gained importance in the field of cardiovascular biology due to its regulatory action on the low-density lipoprotein receptor (LDLR). However, recently, it has also been shown that PCSK9 acts independent of LDLR to cause vascular inflammation and increase the severity of several cardiovascular disorders. We hypothesized that PCSK9 affects the expression of chemokine receptors, major mediators of inflammation, to influence cardiovascular health. However, using overexpression of PCSK9 in murine models in vivo and PCSK9 stimulation of myeloid and vascular cells in vitro did not reveal influences of PCSK9 on the expression of certain chemokine receptors that are known to be involved in the development and progression of atherosclerosis and vascular inflammation. Hence, we conclude that the inflammatory effects of PCSK9 are not associated with the here investigated chemokine receptors and additional research is required to elucidate which mechanisms mediate PCSK9 effects independent of LDLR.

## 1. Introduction

Proprotein convertase subtilisin/kexin type 9 (PCSK9) is a protease mainly secreted by the liver and has been extensively studied as a focal player in lipid metabolism [1]. PCSK9 binds to the low-density lipoprotein receptor (LDLR), causing the lysosomal degradation of the receptor. As this receptor mediates the clearance of LDL-cholesterol (LDL-C), its degradation causes an increase in circulating cholesterol levels [2]. In this context, major efforts over the last years have resulted in the clinical application of anti-PCSK9 antibodies to reduce circulating lipid levels and thereby the risk for cardiovascular disease [3,4,5].

Lately, PCSK9 has also been correlated with having pathological influence on vascular diseases independent of its interaction with the LDLR. Besides hepatocytes in the liver, it has been demonstrated in recent times that many other cells possibly express PCSK9 such as macrophages [6,7,8], cardiomyocytes [9,10], vascular endothelial, and smooth muscle cells [11,12]. It is becoming increasingly evident that PCSK9 does not only cause vascular pathologies by disturbing cholesterol metabolism, but also influences other potential factors such as inflammation and stress responses to develop and promote numerous diseases [1,13]. For instance, circulating PCSK9 has effects on platelet reactivity [14], Apolipoprotein E receptor 2 (ApoER2) [15], and Cluster of differentiation 36 (CD36) [16] independent of LDLR in causing vascular aging, hypertension, myocardial infarction, and atherosclerosis. In line with this, plasma concentrations of PCSK9 are positively correlated with the production of inflammatory cytokines, and directly produces pro-inflammatory responses on cells in vitro by increasing the secretion of inflammatory cytokines [17]. This aggravated inflammation by PCSK9 is thereby believed to play an important role in cardiovascular complications independent of the LDLR [17].

Cardiovascular inflammation is an important factor to be considered in the development and progression of several cardiovascular complications such as atherosclerosis [18]. Important mediators implicated in the process of inflammation and deterioration of cardiovascular complications are chemokine receptors. Chemokines are the main ligands of chemokine receptors and are a family of small homologous proteins [19] present on the surface of a variety of cells, and aid in the recruitment of neutrophils, monocytes, and lymphocytes to the site of inflammation [20]. Chemokine receptors are classified into different groups, namely CCR, CXCR, CX3CR, and XCR depending on the presence of cysteine residues [21]. Besides their role in leukocyte migration, activation of chemokine receptors also triggers several intracellular signaling cascades as well as extracellular processes including the regulation of inflammation and immunity [22] by influencing cell adhesion, cell differentiation, production of cytokines, and other immune regulators, apoptosis, and angiogenesis [23].

Based on the fact that PCSK9 seems to aggravate inflammation independent of the LDL-R, we hypothesize that PCSK9 influences inflammation by modulating certain chemokine receptors. Therefore, in this study, we aimed to investigate whether PCSK9 influences the expression of selected chemokine receptors as well as cytokines that are well known to participate in atherosclerosis and other cardiovascular complications (CXCR4, CCR8, CCR5, and CCR2) in in vitro and in vivo models, which could unveil a novel approach to modulate vascular inflammation.

## 2. Results

### 2.1. Overexpression of PCSK9 In Vivo Does Minimally Affect Leukocyte Chemokine Receptor Expression, Even though It Affects the Expression of Inflammatory Cytokines

To investigate the effect of PCSK9 on leukocytes, C57BL/6 mice were injected with adeno-associated virus type 8 containing D377Y-mPCSK9 to overexpress PCSK9 in the liver, resulting in high circulating levels of PCSK9 (Appendix A). After one week, the expression in the chemokine receptors CXCR4, CCR8, CCR5, and CCR2, which are all known to play an important role in inflammation and atherosclerosis development [24,25,26,27,28,29,30,31], was analyzed on the surface of various leukocyte subsets in the blood (Figure 1) and spleen (Figure 2) using flow cytometry. Leukocyte subsets were analyzed using the following combination of antibodies staining the surface markers: neutrophils (CD45^+^CD11b^+^CD115^−^Gr1^high^), monocytes (CD45^+^CD11b^+^CD115^+^), classical monocytes (Gr1^high^ monocytes), non-classical monocytes (Gr1^low^ monocytes), T-cells (CD45^+^GR1^−^CD115^−^CD3^+^), and B-cells (CD45^+^GR1^−^CD115^−^B220^+^). Overall, the expression of these chemokine receptors remained rather unaffected by the high circulating levels of PCSK9 in vivo. Two specific changes could be observed, being an upregulation of the CCR8 expression on neutrophils in the circulation, which corresponded with an increased expression in total leukocytes and a downregulation of CCR2 in splenic T cells. Although the observed changes were considerable, the expression of these receptors on the respective cell-type was very low (Appendix A), and therefore it is rather unlikely that these changes result in (patho)-physiological effects.

In order to evaluate the effect of PCSK9 on more resident immune cells, peritoneal macrophages were also isolated from C57BL/6 mice overexpressing PCSK9 as described above and the expression of the four selected chemokine receptors (CXCR4, CCR8, CCR5, and CCR2) was again evaluated using flow cytometry. In line with the effects of PCSK9 on circulating and splenic immune cells, the expression of chemokine receptors did not change in peritoneal macrophages upon systemic PCSK9 overexpression (Figure 3), although a slight non-significant increase could be observed in CCR8 expression.

Furthermore, the expression levels of inflammatory cytokines were analyzed in the plasma of these mice (Figure 4). Although almost all cytokines were not significantly changed (*p*-value ≥ 0.05, using Mann–Whitney test), except for CXCL10 and TNF-α, all were upregulated upon PCSK9 stimulation, suggesting pro-inflammatory effects.

### 2.2. PCSK9 Hardly Changes the Expression of Chemokine Receptors and Cytokines on Macrophages

To check the effect of PCSK9 on macrophages from another source, we used freshly differentiated mouse bone marrow derived macrophages (BMDMs) that were incubated with recombinant PCSK9. No significant effects were observed regarding CCR5 and CCR2 in freshly differentiated mouse BMDMs that are incubated with recombinant PCSK9, though a small but significant reduction in CCR8 and CXCR4 expression could be observed in the absence of an inflammatory stimulus (Figure 5A). However, PCSK9 did not affect chemokine receptor expression in BMDMs that are activated using lipopolysaccharide (LPS) to reflect an inflammatory environment (Figure 5B). PCSK9 also failed to influence the expression of the inflammatory cytokines IL6 and TNF-α (Figure 5C,D). Additionally, the effect of PCSK9 has also been investigated using human monocyte derived macrophages to verify that the lack of effect of PCSK9 on chemokine receptor expression is not species restricted. In line with the effects observed in mouse macrophages, PCSK9 did not have any effect on chemokine receptor expression in human macrophages (Figure 6A,B), although a slight non-significant decrease in CXCR4 expression could be observed in the baseline condition. Furthermore, PCSK9 did not have an effect on the expression of inflammatory cytokines (Figure 6C,D).

### 2.3. Chemokine Receptor Expression on Vascular Cells Is Barely Influenced by PCSK9

As vascular cells play a prominent role in the process of development and progression of inflammation, the effect of PCSK9 on chemokine receptor expression was also investigated in these cells. Human aortic smooth muscle cells (HAoSMCs) and human coronary artery endothelial cells (HCAECs) were cultured in vitro and incubated with recombinant PCSK9 in the presence and absence of LPS as an inflammatory trigger. Although a slight significant increase in CXCR4 expression was observed in HCAECs upon PCSK9 treatment, most observations in these vascular cells were in line with those made in hematopoietic cells where PCSK9 hardly influenced the chemokine receptor expression (Figure 7A,B). To determine whether the increase in CXCR4 expression was related to a specific stimuli, we exposed HCAECs to oxidized low-density lipoprotein (oxLDL) and Tumor Necrosis Factor-alpha (TNF-α) in the presence and absence of PCSK9. Although the expression of CXCR4 very slightly, though significantly increased in the presence of TNF-α (Appendix A), we observed that there was no difference in the expression of the receptor in the presence of oxLDL (Appendix A). In line with other cell types, there was also no significant change in the expression of chemokine receptors in the presence of PCSK9 in HAoSMCs (Figure 8A,B), either with or without an inflammatory stimulus, although small non-significant changes in expression could be observed.

However, an interesting effect on the production of inflammatory cytokines was observed. PCSK9 increased the production of IL6 in HCAECs (Figure 7C), while this was not the case in HAoSMCs (Figure 8C). The expression of TNF-α (Figure 7D and Figure 8D) remained unaffected by PCSK9 in both HCAECs and HAoSMCs, reflecting a certain degree of specificity.

## 3. Discussion

It has already been well recognized that PCSK9 plays an important role in CVDs [1,13]. Therefore, over the last years, several therapeutic approaches have been developed to inhibit PCSK9, more particularly the binding of PCSK9 with the LDLR [32]. For example, it has been shown that the monoclonal antibodies Alirocumab and Evolocumab reduce the risk of CVDs on top of standard statin treatments, demonstrating a clear additional effect of PCSK9 targeting [3,33]. Several recent studies have highlighted that PCSK9 can also have various additional effects that are independent of its binding to LDLR [34,35]. For example, PCSK9 can directly modulate inflammatory pathways and stress responses independent of LDLR [17]. Vice-versa, inflammatory pathways such as interferon (IFN)-γ can also increase both PCSK9 gene and protein expression [13]. In line with this, it could also be observed that the inflammatory pathways of STAT3 [36] and SOCS3 [37] induce the expression of PCSK9, clearly demonstrating that PCSK9 expression increases in an inflammatory environment. Additionally, it could be shown that PCSK9 aggravates atherosclerosis in apolipoprotein-E deficient mice [38], which was associated with an increased expression of inflammatory cytokines such as TNF-α, IL-1β, IL-10, MCP-1, and IL-8 to directly facilitate the process of inflammation. Further confirming this, inhibition of PCSK9 was shown to reduce the recruitment of inflammatory monocytes to atherosclerotic lesions [39]. Taking into account the fact that PCSK9 seems to be directly involved in the regulation of inflammatory processes, independent of LDLR, we reasoned that PCSK9 might also influence the expression of chemokine receptors, which play an important role in various pathologies.

In cardiovascular disorders such as atherosclerosis and myocardial ischemia, endothelial cell injury leads to the expression of inflammatory cytokines, resulting in leukocyte recruitment. Chemokines and chemokine receptors play a vital role in this migration and trafficking of hematopoietic cells to the sites of injury. For example, CCR2 is one of the chemokine receptors that has been shown to play a key role in the development of CVDs in mice [40] and human vascular tissues [24,41] and serves as a marker for inflammation [42]. In sites of inflammation, CCR2 is overexpressed to recruit hematopoietic cells [42] and therefore critically involved in the initial inflammatory responses. A study observed that patients with hypercholesterolemia showed increased expression of CCR2 on classical monocytes [42], but treating them with PCSK9 inhibitors significantly lowered the expression of the same receptor. In this case, the monoclonal antibodies (mAbs) that inhibit the PCSK9-LDLR binding also decreased the monocyte CCR2 expression. In contrast, another study treated monocytes with statins that increases PCSK9 levels, but decreases HMG-CoA reductase levels and observed that CCR2 expression on the monocytes was reduced [43]. From these studies, it could be hypothesized that CCR2 expression might be influenced by the presence of PCSK9, even though it is debatable whether it upregulates or downregulates it. In contrast to these observations, our experiments showed that in vitro or in vivo, overexpression of PCSK9 did not change the expression of CCR2 on the surface of most immune or vascular cells, although in HAoSMCs at baseline, a small decrease in expression could be observed. Furthermore, we were able to see a reduction in splenic T cell CCR2, which is a cell-type with a low expression of CCR2 that is not (patho)physiologically relevant. It might be possible that this discrepancy is based on the exact investigated cell type, as we did not investigate the effects of CCR2 on freshly isolated human monocytes. Further studies are needed to further pinpoint the exact cause of these apparent discrepancies.

Various studies have observed that inhibition of CCR5 reduces the production of inflammatory cytokines [44,45] and atherosclerosis [46]. Therapeutic targeting of CCR5 facilitates the control of inflammatory situations, especially in the context of cardiac complications. Similarly, the complex of CCR8 with its ligands is believed to be upregulated in various inflammatory diseases. Besides a small effect of PCSK9 on CCR8 in peritoneal macrophages, no obvious differences were observed upon PCSK9 treatment, indicating that PCSK9 does not influence inflammatory processes through these chemokine receptors. The receptor CXCR4 has also been extensively studied in various inflammatory situations. It plays, for example, a part in modulating cell chemotaxis [47], apoptosis [48] as well as cell survival [49,50]. The transcription of this receptor is believed to increase in the presence of the various cytokines that include IL-2, IL-4, IL-7, IL-10, IL-15, and TGF-β, whereas in the presence of cytokines such as TNF-α, INF-γ, and IL-1β, the expression of CXCR4 diminishes [51]. CXCR4 knockout in cardiomyocytes leads to progressive cardiac dysfunction and later to cardiac failure [52]. Similarly knocking out the receptor in endothelial cells and smooth muscle cells aids in the progression of atherosclerosis [53,54]. All of the studies on CXCR4 and the vascular system substantiates the notion that the expression of CXCR4 provides a protective function against several CVDs. In our study, treatment with PCSK9 failed to regulate the expression of CXCR4 in hematopoietic cells, although in human macrophages, a small non-significant decrease in expression could be observed at baseline. Interestingly, PCSK9 seems to result in a rather limited increase of the expression of CXCR4 in HCAECs, which would be in sharp contrast to the generally described protective function of CXCR4 in the vasculature. Further studies are needed to validate whether this increased expression would be sufficient to result in any functional differences regarding endothelial cell phenotype.

In our study, we observed that in C57/BL6 mice in vivo, PCSK9 influences the expression of certain inflammatory cytokines such as CXCL10 and TNF-α. However, in vitro PCSK9 does not affect the expression of IL-6 or TNF-α in smooth muscle cells or macrophages. We were able to observe a significant increase in the expression of IL-6 by endothelial cells in the presence of PCSK9, suggesting that PCSK9 does mediate inflammation, especially in these cells. It would be interesting to evaluate other inflammatory cytokines and mediators to understand different pathways through which PCSK9 might impact endothelial inflammation.

Recent studies have clearly identified that there is a bi-directional interaction between PCSK9 and inflammatory mediators, which could thereby, in a LDLR independent manner, contribute to cardiovascular effects of PCSK9. We are the first to investigate whether this interaction also involves chemokine receptors, which at least for the four important receptors that we have studied does not seem to be the case. Although minor changes are observed in specific cell-subsets, overall PCSK9 does not seem to exert detrimental effects via modulation of chemokine receptor expression. As our in vivo murine studies were performed on chow diet, future experiments investigating the effects of PCSK9 on chemokine receptors in atherosclerotic mouse models will also be interesting. Therefore, future studies should further investigate which other inflammatory mediators are influenced by PCSK9. Further pinpointing interesting interactions of PCSK9 with other inflammatory mediators would be of high importance to fully understand the mechanism by which PCSK9 affects CVDs.

## 4. Materials and Methods

### 4.1. In Vitro Culture and Cell Treatment

Bone marrow was isolated from femurs and tibiae of C57BL/6 mice (Janvier Labs) and cultured in RPMI-1640 medium (Gibco by Life technologies) supplemented with 10% (vol/vol) heat inactivated fetal calf serum, L-glutamine, and 1% penicillin/streptomycin (All Gibco by Life technologies) and 25 ng/mL recombinant murine monocyte colony stimulating factor (M-CSF; PeproTech Inc., Merseburg, Germany) for 8–9 days to differentiate into bone marrow-derived macrophages (BMDMs). All animal experiments were approved by the local authorities (Landesamt für Natur, Umwelt und Verbraucherschutz Nordrhein-Westfalen, Germany, approval number 40097A4) and complied with the German animal protection law.

Peripheral blood mononuclear cells (PBMCs) were isolated from buffy coats from healthy donors provided by the blood bank at Uniklinikum RWTH Aachen, and in accordance with the regulations on the secondary use of human tissue and in accordance with the guidelines of the local medical and ethical committee. PBMCs were incubated with anti-CD14 beads and passed through a MACS separator (Milteny Biotech) to separate monocytes from other leukocyte sub-types. These monocytes were incubated for seven days with 500 ng/mL recombinant human macrophage colony stimulating factor (M-CSF) purchased from PeproTech Inc. to enable differentiation of monocytes into macrophages. Cryopreserved human coronary artery endothelial cells from Lonza (catalog#: CC-2585) and human aortic smooth muscle cells from Promocell (catalog#: C-12533) were purchased and cultured according to the manufacturer’s instructions.

All cells were first pre-stimulated with 5 µg/mL of human recombinant PCSK9 (Biolegend) for 24 h and then washed and stimulated with 10 ng/mL lipopolysaccharide (LPS; Sigma Aldrich, Darmstadt, Germany) for 6 h to create an inflammatory environment (Appendix A).

### 4.2. In Vivo Mouse Model

C57BL/6 mice (Janvier Labs) were injected i.v. with either a recombinant adenoviral construct AAV8-D337Y-PCSK9 from Vector Biolabs at 1 × 10^11^ gc concentration, ensuring overexpression of PCSK9 [55] (*n* = 8) or an empty AAV8 as the control (*n* = 8). Thereafter, the animals were kept on a chow diet for one week and subsequently euthanized to extract blood and organs. All animal experiments were approved by the local authorities (Landesamt für Natur, Umwelt und Verbraucherschutz Nordrhein-Westfalen, Germany, approval number 81-02.04.2019.A363) and complied with the German animal protection law.

### 4.3. Flow Cytometry

Whole blood obtained from the retro-orbital plexus of mice was EDTA-buffered and subjected to red blood cell lysis. Isolated spleens from these mice were smashed and passed through 30 µm filters to collect the cells, which were subsequently subjected to red blood cell lysis. Leukocyte subsets were analyzed with antibodies purchased from Abcam with a concentration of 0.5 µg using the following combination of surface markers: neutrophils (CD45^+^CD11b^+^CD115^−^Gr1^high^), monocytes (CD45^+^CD11b^+^CD115^+^), classical monocytes or inflammatory monocytes (Gr1^high^ monocytes), non-classical monocytes or anti-inflammatory monocytes (Gr1^low^ monocytes), T-cells (CD45^+^GR1^−^CD115^−^CD3^+^), and B-cells (CD45^+^GR1^−^CD115^−^B220^+^). Resident peritoneal macrophages were collected by flushing the peritoneal cavity with ice-cold PBS, and were subsequently analyzed using flow cytometry for the expression of chemokine receptors. Flow cytometry antibodies conjugated with specific fluorophores were purchased from Abcam, namely: CD45—APC Cy7, CD115—FITC, CD11b—V500, GR1—PE Cy7, CD45R/B220—eF450, CD3—PerCP, CXCR4—APC, CCR5—APC, CCR8—PE, and CCR2—PE. All the cell populations were stained with antibodies with a concentration of 0.5 µg for CCR2, CCR5, CCR8, and CXCR4 purchased from Abcam. The histograms of all the receptors investigated stained with the above-mentioned antibodies compared to the unstained leukocytes are shown in Figure 1A. Cell populations and marker expression were analyzed after appropriate compensation and gating using FACSCanto-II FACSDiva software (BD Biosciences) and the FlowJo analysis program by Becton Dickinson and Company.

### 4.4. ELISA

Inflammatory cytokines (TNF-α and IL6) were measured using enyzme-linked immunosorbent assays (ELISA) from ThermoFisher Scientific according to the manufacturer’s protocols.

### 4.5. LUMINEX

Commercially available Luminex multiplex immuno assay with superparamagnetic beads from R&D systems was used and the expression of inflammatory cytokines in the plasma of C57/Bl6 mice and C57/BL6 mice overexpressing PCSK9 were measured using Luminex MAGPIX instruments from R&D systems according to protocols provided by the manufacturer.

### 4.6. PCSK9 ELISA

Mouse PCSK9 ELISA Kits (Abcam) were used to analyze the levels of PCSK9 in the plasma of C57/BL6 mice that were either injected with AAV8-D337Y-PCSK9 or an empty AAV8 as the control using protocols provided by the manufacturer.

### 4.7. Statistical Analysis

Statistical analysis was performed using parametric t-tests or 2-way ANOVA for graphs with normal data distribution and the Mann–Whitney test for graphs with data groups that did not pass D’Agostino and Pearson normality tests. Data were expressed as mean ± SEM. Statistical significance was set at 0.05.

## Figures and Tables

**Figure 1 ijms-22-13026-f001:**
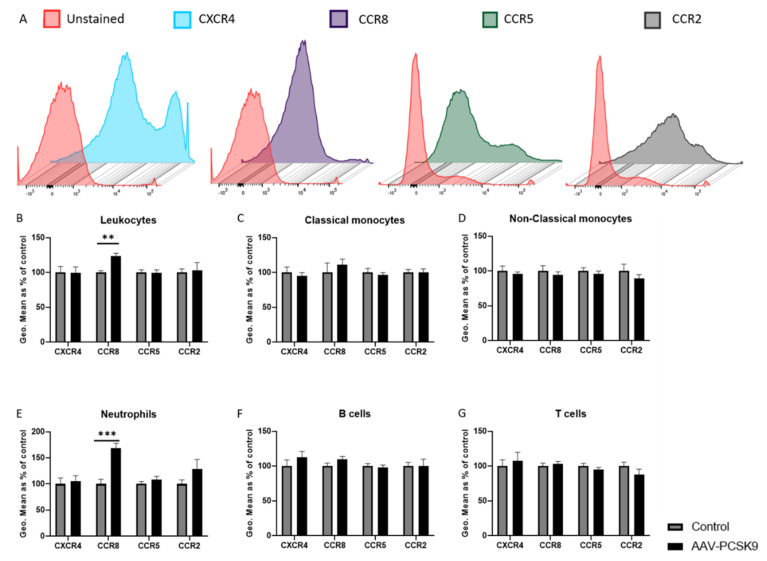
**PCSK9 only slightly affects the expression of chemokine receptors on blood leukocytes and subtypes in vivo.** The expression of antibodies used for CXCR4, CCR8, CCR5 and CCR2 were tested on leukocytes in the blood of C57/BL6 mice and compared to unstained leukocytes to represent the efficiency of the antibodies (**A**). The expression levels of chemokine receptors was measured using flow cytometry and represented using geometric mean (Geo.Mean) in blood leukocytes (**B**), classical monocytes (**C**), non-classical monocytes (**D**), neutrophils (**E**), B cells (**F**) and T cells (**G**) of PCSK9 over-expressing mice (*n* = 8) and compared to control mice (*n* = 8). Bar graphs are representation of mean ± SEM. ** *p* < 0.01, *** *p* < 0.001. The statistical difference between the groups was calculated using 2-way ANOVA tests in GraphPad Prism.

**Figure 2 ijms-22-13026-f002:**
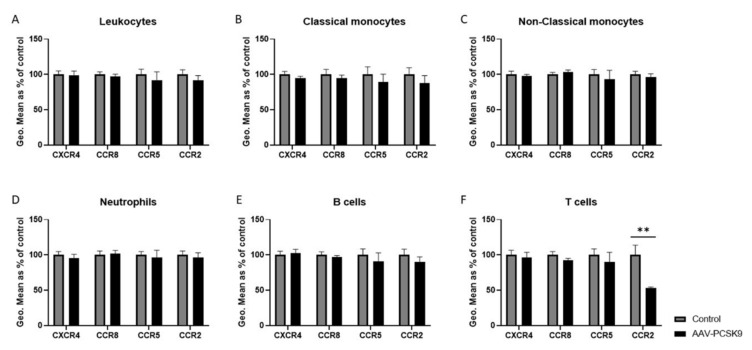
**The expression of chemokine receptors on splenic leukocytes is not affected by PCSK9 overexpression in vivo, except in T-cells.** The expression of chemokine receptors was measured using flow cytometry and represented using geometric mean (Geo.Mean) in splenic leukocytes (**A**), classical monocytes (**B**), non-classical monocytes (**C**), neutrophils (**D**), B cells (**E**), and T cells (**F**) of PCSK9 over-expressing mice (*n* = 8) and compared to control mice (*n* = 8). Bar graphs are a representation of mean ± SEM. ** *p* < 0.01. The statistical difference between the groups was calculated using 2-way ANOVA tests in GraphPad Prism.

**Figure 3 ijms-22-13026-f003:**
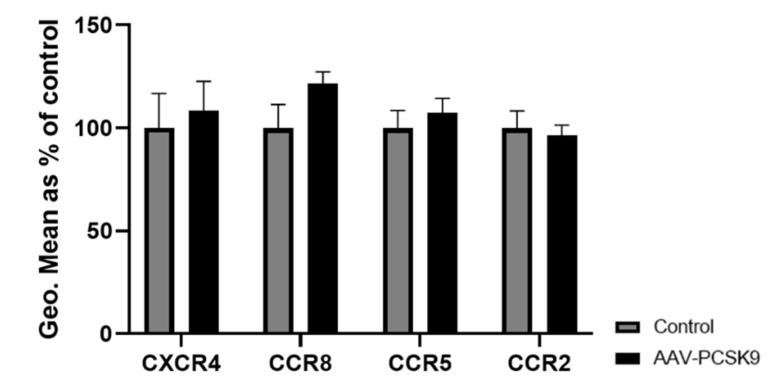
**PCSK9 does not change the expression of chemokine receptors in peritoneal macrophages in vivo.** The expression of chemokine receptors was measured using flow cytometry, represented using geometric mean (Geo.Mean) in peritoneal macrophages isolated from PCSK9 over-expressing mice (*n* = 8) and compared to control mice (*n* = 8). Bar graphs are representation of mean ± SEM. The statistical difference between the groups was calculated using 2-way ANOVA tests in GraphPad Prism.

**Figure 4 ijms-22-13026-f004:**
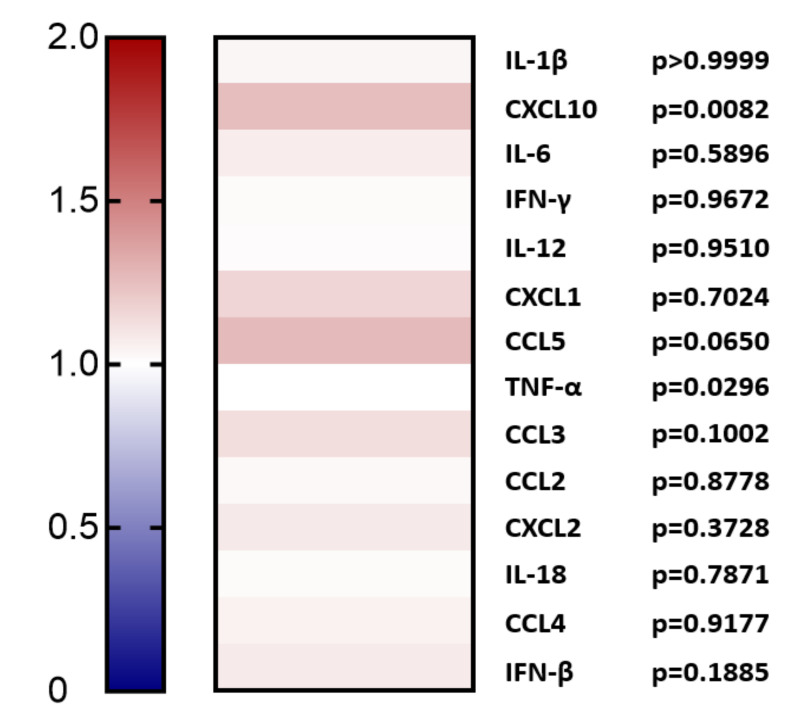
**Heat map analysis of the expression of cytokines in the plasma of mice over-expressing PCSK9 in vivo in comparison to control mice.** The expression of inflammatory cytokines was measured using the Luminex assay in the plasma of the PCSK9 over-expressing mice (*n* = 8) and compared to the control mice (*n* = 8). Data are represented in the form of fold change for upregulated and downregulated cytokines with respect to the control mice. Mann–Whitney test was performed to analyze the disparities between the groups, and the groups with *p* < 0.05 were considered significantly different.

**Figure 5 ijms-22-13026-f005:**
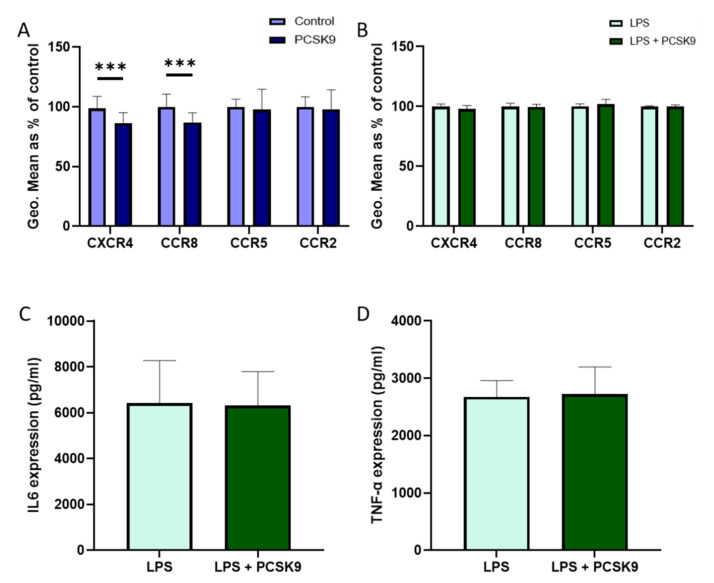
**PCSK9 does not change the expression of chemokine receptors and inflammatory cytokines in mouse bone-marrow derived macrophages in vitro.** The expression of chemokine receptors CXCR4, CCR8, CCR5, and CCR2 were measured using flow cytometry and represented using geometric mean (Geo.Mean) in mouse bone-marrow derived macrophages (*n* = 3–4 replicates) with and without PCSK9 (**A**) and stimulated with LPS (**B**). Inflammatory cytokines IL6 (**C**) and TNF-α (**D**) expression measured in the supernatant of the cell culture with and without PCSK9 in an inflammatory environment (*n* = 3–4 replicates). Bar graphs are representation of mean ± SEM. *** *p* < 0.001. 2-way ANOVA was used for (**A**,**B**), and Mann–Whitney test was used for (**C**,**D**) to calculate the statistical differences.

**Figure 6 ijms-22-13026-f006:**
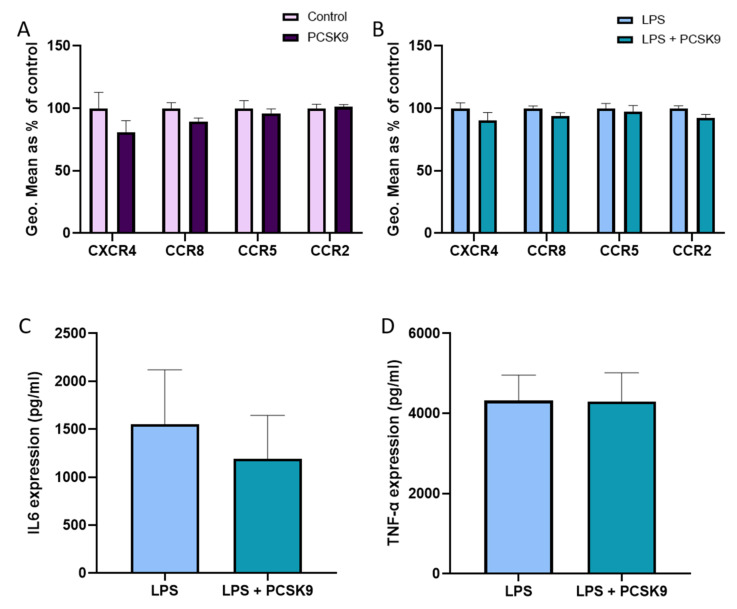
**PCSK9 does not change the expression of chemokine receptors and inflammatory cytokines in human macrophages in vitro.** Peripheral blood mononuclear cells (PBMCs) isolated from buffy coats were incubated with monocyte-specific antibodies to isolate human monocytes. These monocytes were then cultured with macrophage colony stimulating factor (M-CSF) for seven days, allowing them to differentiate into macrophages. The expression of chemokine receptors CXCR4, CCR8, CCR5, and CCR2 measured using flow cytometry and represented using geometric mean (Geo.Mean) in human macrophages (*n* = 3–4 replicates) with and without PCSK9 (**A**) and when exposed to inflammatory stimulus with LPS (**B**). Inflammatory cytokines IL6 (**C**) and TNF-α (**D**) expression measured in the supernatant of the cell culture with and without PCSK9 in an inflammatory environment (*n* = 3–4 replicates). Bar graphs are representation of mean ± SEM. 2-way ANOVA was used for (**A**,**B**), and Mann–Whitney test was used for (**C**,**D**) to calculate the statistical differences.

**Figure 7 ijms-22-13026-f007:**
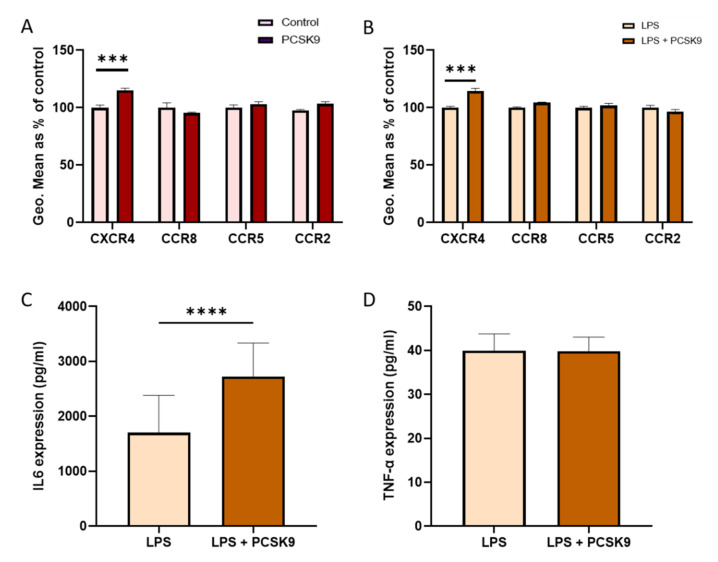
**PCSK9 marginally affects the expression of chemokine receptors in human coronary artery endothelial cells while considerably increasing the secretion of IL6.** The expression of chemokine receptors CXCR4, CCR8, CCR5, and CCR2 were measured using flow cytometry and represented using geometric mean (Geo.Mean) in HCAECs (*n* = 3–4 replicates) with and without PCSK9 (**A**) and when exposed to an inflammatory environment with LPS (**B**). Inflammatory cytokines IL6 (**C**) and TNF-α (**D**) expressions measured in the supernatant of the cell culture with and without PCSK9 in an inflammatory environment (*n* = 3–4 replicates). Bar graphs are representation of mean ± SEM. *** *p* < 0.001, **** *p* < 0.0001. 2-way ANOVA was used for (**A**,**B**), and the Mann–Whitney test was used for (**C**,**D**) to calculate the statistical differences.

**Figure 8 ijms-22-13026-f008:**
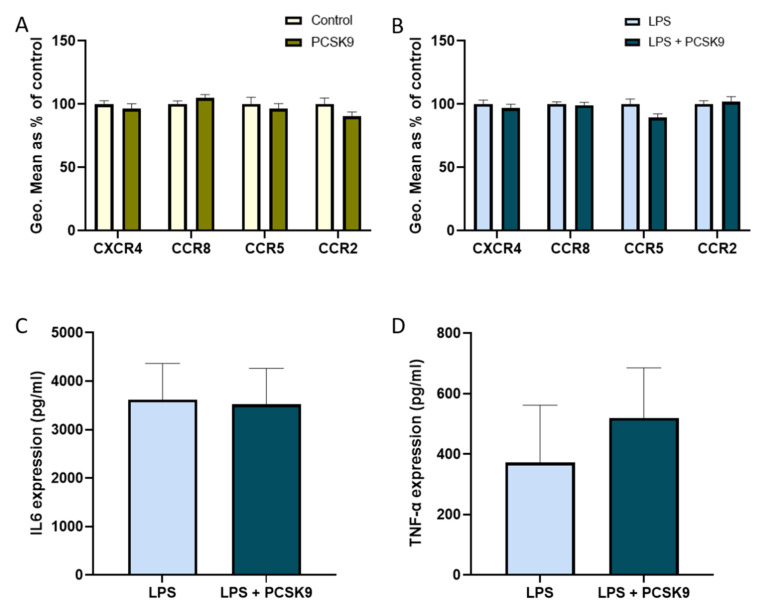
**PCSK9 has no effect on the chemokine receptors and cytokines in human aortic smooth muscle cells.** The expression of chemokine receptors CXCR4, CCR8, CCR5, and CCR2 were measured using flow cytometry and represented using geometric mean (Geo.Mean) in HAoSMCs (*n* = 3–4 replicates) with and without PCSK9 (**A**) and when exposed to an inflammatory environment with LPS (**B**). Inflammatory cytokines IL6 (**C**) and TNF-α (**D**) expressions measured in the supernatant of the cell culture with and without PCSK9 in an inflammatory environment (*n* = 3–4 replicates). Bar graphs are representation of mean ± SEM. 2-way ANOVA was used for (**A**,**B**) and thee Mann–Whitney test was used for (**C**,**D**) to calculate the statistical differences.

## Data Availability

Not applicable.

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
