# Peer review of "PCSK9 Imperceptibly Affects Chemokine Receptor Expression In Vitro and In Vivo"

_ijms, 2021, doi:10.3390/ijms222313026_

Round 1

Reviewer 1 Report

The authors of manuscript titled “PCSK9 imperceptibly affects chemokine receptor expression in-vitro and in-vivo” hypothesized that PCSK9 will affect the expression of chemokine receptors. With the use of selected immune cell types and expression levels of chemokine receptors or cytokines, the authors concluded that PCSK9’s inflammatory effects are independent of the expression levels of chemokine receptors CXCR4, CCR8, CCR5 and CCR2.

Structurally the study and results are displayed logically and clear, but there are major items in the methodology that needs to be addressed so the value of what was done can be fully appreciated.

For example,

  1. Much of the conclusions of the study and discussion are dependent on the statistical analysis of the data on the expression of the chemokine receptors but the utility of the statical analysis seems not standardized despite the description in the methods.

Without raw data and by use of the eye, Fig3 CCR8 is seen upregulated, CXCR4 and CCR4 downregulated, Figure 6 CCR downregulated, and Fig 8 A. CCR2 downregulated and B. CCR5 downregulated. With analyzing all data or demonstrating exact method used to determine statical significance of each figure, updated description of the results and discussion should follow.

  1. Figure 4 legend does not define the p value.
  2. Supplementary Figure 4 is completely missing.
  3. Definition of Geo. Mean as …… in legends are commonly missing
  4. Where stated with in-vitro cell lines (n=3-4), what does this mean exactly? 3-4 cells or times that experiments were performed.
  5. Generally how many times were the experiments performed? Or replicated?
  6. In methods section how many control mice and tested mice?
  7. Ethical statement for the PBMCs study is needed.
  8. Section 4.1’s mice approval: is it the same as the approval for section 4.2?
  9. Line 339: As most figures rely on accurate isolation (analytical or physical) of leukocytes, more information is need on how the Leukocyte subsets were analyzed? With antibodies, kit??? Citation??
  10. Line 345: All cell population were stained with antibodies… What dilution of the antibody, what is the secondary antibody? And these cells were immunolabelled correct? Not stained, or am I missing something in the method here as it is not described exactly.
  11. Line 265: Contents of PCSK9. This is misleading. What type of kit was it? An ELISA or Western BLOT or something else.
  12. Which figures did you use Ordinary one-way ANOVA and which for Kurskal-Wallis test.
  13. You set Statistical significance was set at 0.05 but in the figure legends of some it is not so.
  14. Is there a citation for the AAV vectors and their use of them?
  15. What is meant by non-classical monocytes?

Results

-Most figures have text that are pixelated.

-Line 132 starting with Although….not significantly changed needs more clarity ie. Based on p = 0.05 or

-Lines 165 and 172 and 200 and 210: legends state inflammatory environment but does not clarify this in the methods section with the use of LPS. Authors want the readers to assume that LPS or PCSK9 causes an inflammatory response is said cell lines, but there is no evidence or data that demonstrates this in this manuscript. Please provide the data or at least a citation for use of LPS and PCSK9 alone (and its amount) in creating an inflammatory response in these cell lines. These are nice controls for such experiments, as are the cell lines alone to see basal levels of IL6 and TNFalpha in these cell lines.

-Clarity of the differentiated macrophages from PBMCs in the figure legend is needed.

Language and manuscript

Parts of the text is difficult to follow and suggest rewording the sentences:

-Line 29 Sentence starting with Hence: independent of, are independent, independent of, and  of the here investigated chemokine, further and further……

-Line 41: in the circulating change to in circulating

-Line 47: Besides the hepatocytes change to Besides hepatocytes

-Line 48: in the recent times change to in recent

-Line 51: the cholesterol metabolism to cholesterol metabolism

-Line 55: to cause change to in causing

-Line 122: did also not change to did not

Other

-Spacing in manuscript is not consistent; i.e. line 83 to Line 84 and Line 211 to 214

-In-vitro and In-vivo should be italicized and without – but a space as in: in vitro

-25ng/ml should be 25 ng/ml. Make changes throughout the manuscript for this spacing issue and including 30mm to 30 mm.

Reviewer 2 Report

The issue as to whether PCSK9 impacts on inflammation or the other way around is still debated. The present findings are interesting, but discussion should be more focalized. The authors missed to mention that PCSK9 is raised by the inflammatory pathways of SOCS3 and STAT3, as well as by interferon gamma (PMID: 30477320). A recent review on the pleiotropic effect of PCSK9 could be quoted in the introduction (PMID: 34019847). PCSK9 inhibitors do not reduce inflammation.

Data on LPS and PCSK9 were already published. Which is the plus of this experiment? Figure legends should report the statistical analysis that has been used. Please be consistent with the use of coloured pictures throughout the manuscript.

Did mice overexpressing PCSK9 developed a plaque? (PMID: 22261195)

Round 2

Reviewer 1 Report

Well done with the improvements and it is easier to read and the impact can be valued more. One very small spacing issue: line 397 at 

"aconcentration" should be a concentration.

And Figure S5 could be in black and white to be presented like the other Supplementary figures.

Reviewer 2 Report

No further comments